# Vitreous Foam with Thermal Insulating Property Produced with the Addition of Waste Glass Powder and Rice Husk Ash

**Fernando Antonio da Silva Fernandes** [1,2,*], **Dayriane do Socorro de Oliveira Costa** [3],
**Camilo Andrés Guerrero Martin** [4] **and João Adriano Rossignolo** [2]

1   Department of Biosystems Engineering, University of São Paulo, Av. Duque de Caxias Norte, 225, Pirassununga 13635-900, SP, Brazil
2   Department of Engineering, Federal University of Pará—Campus Salinópolis, Rua Raimundo Santana Cruz, S/N, Bairro São Tomé, Salinópolis 68721-000, PA, Brazil
3   Department of Engineering, Federal University of Rio de Janeiro, Rio de Janeiro 21941-853, RJ, Brazil
4   Laboratório de Operações e Tecnologias Energéticas Aplicadas na Indústria do Petróleo, Faculty of 7 Petroleum Engineering, Federal University of Pará, Salinópolis 68721-000, PA, Brazil
*   Correspondence: fernandesfernando27@gmail.com; Tel.: +55-63-98433-0565

**Abstract:** Closed pore glass foams with dimensions of 60mm $\times$ 20 mm $\times$ 20 mm were produced using agro-industrial residues. Samples containing sodo-calcic glass powder (78%wt) and rice husk ash (16%wt) were characterized, and their technological properties were investigated. The samples were synthesized in a conventional muffle furnace at 750–800–850 °C. The results presented for apparent density (0.24–0.29 g/cm$^3$), compressive strength (1.5–2.3 MPa) and thermal conductivity (0.021–0.025 W/mK) meet the standards for commercial foam glasses. Low viscosity was achieved at all temperatures as a result of the addition of rice husk ash to the cell structure. X-ray fluorescence showed that the glass was silico-sodo-calcic type (SiO$_2$, Na$_2$O and CaO), and that the rice husk ash was rich in SiO$_2$ (as well as CaO, Na$_2$O, Al$_2$O$_3$, K$_2$O and Fe$_2$O$_3$). The mechanical strength and low thermal conduction of the material showed a good efficiency for use in civil construction as a thermal insulating material. Material made in this way has a lower production cost, and additionally transforms waste into co-products, generating added value, favoring consecutive circulation, as well as a clean and circular economy.

**Keywords:** agro-industrial waste; glass foam; thermal insulation; circular economy

## 1. Introduction

The rapid expansion of cities around the world, caused by population growth has resulted in increased resource consumption and waste production [1]. Civil construction is always involved in discussions about sustainability through the use, production and research of new materials [2]. The increase in temperature in several countries, has made the construction sector increase its demand for materials to be used in thermal insulation as well [3]. Eco-design has great interest in the use of materials with insulating properties to ensure energy efficiency in environments during the use phase; as an example, internal and external walls of buildings coated with thermal insulating materials can reduce energy costs in large urban centers [4].

The Circular Economy (EC) has been gaining strength, which increases the focus on the development of circular buildings, requiring a greater availability of circular materials and products. Currently, the availability of circular, sustainable and recyclable insulation materials is considered deficient, showing the need for research and development in this direction [4]. This study is in concomitance and corroboration with the promotion of the circular economy that is currently being carried out worldwide [4], and with the correct destination of the waste used as raw material in this study. As an example of relevant research about this theme, it is possible to mention the following studies: Composition

Component Influence on Concrete Properties with the Additive of Rubber Tree Seed Shells [5]; Normal-Weight Concrete with Improved Stress–Strain Characteristics Reinforced with Dispersed Coconut Fibers [6]; Improvement in Bending Performance of Reinforced Concrete Beams Produced with Waste Lathe Scraps [7]; Performance Assessment of Fiber-Reinforced Concrete Produced with Waste Lathe Fibers [8]; Performance evaluation of fiber-reinforced concrete produced with steel fibers extracted from waste tire [9]; Use of recycled coal bottom ash in reinforced concrete beams as replacement for aggregate [10]; A Review on the Effect of Mechanical Properties and Durability of Concrete with Construction and Demolition Waste (CDW) and Fly Ash in the Production of New Cement Concrete [11]; Permeability of recycled aggregate concrete containing fly ash and clay brick waste [12]; Effect of curing temperature and glass type on the pozzolanic reactivity of glass powder [13]; and Reuse of wood ash from biomass combustion in non-structural concrete: mechanical properties, durability, and eco-efficiency [14].

Another important study was regarding the production of glass foam from a mixture of glass powder, rice husk ash (RHA) and calcium carbonate ($CaCO_3$) [15], which attracted great interest in the sector of civil construction, due to its technological features such as being chemically inert, non-toxic, of low density, high in compressive strength, a good thermal insulator and with better weather resistance compared to polymeric foams [4].

Closed-pore glass foams are produced by the powder metallurgy method [16,17], which controls the burning conditions in order to obtain partial densification [18]. Combining residues with raw materials that present ceramic and processing characteristics, closed-pore glass foams present high mechanical strength and structural uniformity [18]. When homogenized and added to the raw materials, calcium carbonate ($CaCO_3$) works as a binder that contributes to melting during burning [19], and also contributes to increasing the strength of the structure after drying, thus preventing collapse during the volatilization of the organic part of the material [18]. The decomposition of $CaCO_3$ takes place between 600 °C and 720 °C, generating CaO and CO [20,21].

In addition, the burning temperature influences the production of vitreous foam [15], and the formation of pores is related to low heating rates [22]. Up to 97% of the vitreous foam mass can be soda-lime glass powder [15,23]. According to the project and its application, production parameters can be modified and adapted, such as the formation technique, gas formed agent and its quantity, etc. [24]. The softened glass mass incorporates the RHA, and guarantees the technological properties of the vitreous foam that serves the civil construction industry [15,25,26]. Studies show that glass waste can be added to concrete to replace cement, replace coarse and fine natural aggregates, and contribute to increasing compressive strength [27–29].

Therefore, from a sustainability point of view, using glass foam in civil construction is a good alternative [30–32]. Incorporating agro-industrial waste in the process of manufacturing vitreous foam as a raw material favors its production, because less energy is used when compared to conventional materials [33]. The agro-industrial residues are incorporated in the glass matrix without compromising its formation, especially residues that present $SiO_2$. RHA features low permeability, high carbonation resistance, chloride resistance, sulfate resistance, and resistance to acidic environments [26]. Waste has been gaining importance due to its properties, such as waste tires [34], agricultural waste [30], concrete waste [35,36], glass waste [3,37] and waste marble powder [38]. A residue that has been gaining importance due to its properties is rice husk, which represents up to 22% of the total mass of rice [3,39,40]. The most produced food in the world is rice, and it is a crop that that generates the most waste (husks) [39]; in 2020, the estimated waste production was 450 million tons [15]. Rice husk ash is the result of burning rice husks during energy (heat) production [26]. RHA represents ±20% of the grain mass, being rich ($SiO_2$) with technological potential for the production of thermal insulating materials [41–44]. The shell burning temperature can produce an ash with an amorphous or crystalline structure [26].

This study produced vitreous foams using glass powder and rice husk ash as raw materials, and the foams were fired (750–800–850 °C). The production of foams was con-

sidered economically viable, because the burning took place at low temperatures, with discarded material was used as raw material. The efficiency of recycled glass and rice husk ash in the production of this material can contribute to promoting the circular economy, and provides used waste a new disposal method, transforming it into co-products.

## 2. Materials and Methods

### 2.1. Material Preparation

The vitreous foams in the green state were produced using glass powder, rice husk ash, calcium carbonate, water and PVA. The glass bottles were washed under running water, and dried at room temperature. Then, they were manually crushed with a hammer to size <10 mm, and all material was crushed using an electric ball mill (TS RUBENS, Brazil) with a speed of 100 rpm until a particle size of $D_{90}$ = 74 µm was reached. Residual RHA presented a particle size of $D_{90}$ = 71 µm, after firing red ceramic pieces in an industry located in the city of Cristalãndia/TO/BRAZIL (10°36′14.7″ S; 49°11′56.0″ W). Industrial $CaCO_3$ was purchased from Dinâmica/Brasil and presented a size of $D_{90}$ = 2 µm. The results of the chemical composition of each raw material were obtained using X-ray fluorescence (XRF, 1800, Shimadzu, São Paulo, Brazil), and are presented in Table 1. The Raman spectroscopic properties of the samples were evaluated (Modelo: Scientific IHR 550, Horiba, Suzhou, China). Figure 1 illustrates the raw materials used and the mixture in the green state.

**Table 1.** Chemical analyses of glass, RHA and $CaCO_3$.

| Material | Composition * | | | | | | |
|---|---|---|---|---|---|---|---|
| | $SiO_2$ | $CaO$ | $Na_2O$ | $Al_2O_3$ | $K_2O$ | $Fe_2O_3$ | $P_2O_5$ |
| Glass | 72.00 | 21.00 | 12.70 | 1.47 | 0.90 | 0.80 | – |
| RHA | 89.00 | 2.70 | 1.70 | 0.97 | 2.70 | 0.33 | 0.98 |
| $CaCO_3$ | 0.41 | 97.80 | – | 0.07 | – | 0.15 | 1.08 |

* Expressed in oxides. MnO, MgO, SrO and $SO_3$ were found in smaller proportions.

**Figure 1.** Rice husk ash, glass bottles; glass powder and the mixture in the green state before firing.

### 2.2. Sample Preparation

The investigation of the technological properties of the foam was carried out on the following samples: soda-lime glass powder (78%wt), rice husk ash (16%wt) and $CaCO_3$ (6%wt) [3,15]. Twelve samples were formulated and investigated at each temperature (750–800–850 °C). The raw materials were classified and weighed, using an analytical balance. Then, they were manually homogenized in a porcelain mortar for 4 min with the addition of water (5%) and PVA solution with 4% of active material (polyvinyl alcohol P.S, Dinâmica Brasil). Polyvinyl alcohol is a material that dissolves in water with a low cost, and shows good agglutination of raw materials [3,45]. Soon after, the material was

placed in a stainless steel die (60 mm × 20 mm × 20 mm) to be pressed by a hydraulic press with a uniaxial load of 40 MPa. At the end of pressing, the samples in the green state were removed from the mold and left to dry at room temperature for 4 h. At the end of drying, they were burned using an electric muffle furnace (EDG, 1800, EDG, Rio de Janeiro, Brazil) at temperatures of 750–800–850 °C, and a heating rate of 100 °C/min for 30 min; this value was defined in the pre-test performed. At the end of the firing, the samples remained inside the muffle until they reached room temperature (cooling); this avoided the appearance of cracks due to the thermal stress that usually accumulates in the cellular structure of the material. To perform characterization by X-ray, a sample burned at each investigated temperature was selected. Figure 2 illustrates the mixture of raw materials used in the green state, pressing, muffle and foam glass.

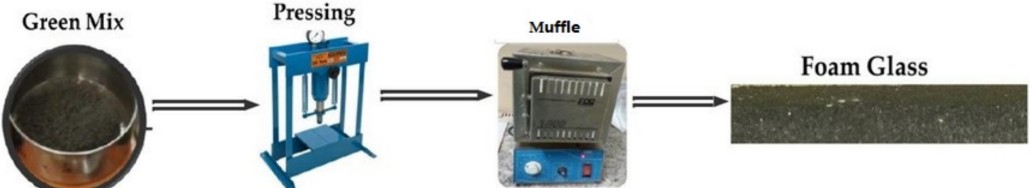

**Figure 2.** Green mix, pressing, muffle and foam glass.

*2.3. Sample Characterization*

Technological properties such as density, apparent porosity, compressive strength and conductivity were evaluated.

$$\varepsilon(\%) = \left[1 - \frac{\rho_a}{\rho_t}\right] \qquad (1)$$

The porosity ($\varepsilon$) was obtained by applying Equation (1), where ($\rho_a$) is the apparent density, and $\rho_t$ = 2.5 g/cm$^3$ is the theoretical density of the powder.

Apparent density was calculated using the mass and geometric dimensions of the samples; this was carried out by Francis and Abdel Rahman [46]. A heating rate of 10 °C/min was applied to perform differential thermal analysis (DTA; SDT–Q6000, TA Instruments) and thermogravimetric analysis (TGA; TGA–50, Shimadzu, São Paulo, Brazil). The identification of the mean distribution of granulometry was carried out (Cilas 1180). The compressive strength was performed with a universal machine (EMIC, DL–3000). To investigate the compressive strength, samples with dimensions of 60 mm × 20 mm × 20 mm with three replications were used. The thermal conductivity of the samples (30mm × 30mm × 30mm) was carried out in accordance with the research of König, J.; Petersen, R.R.; Yue, Y. [47] and da Silva Fernandes, F.A.; de Oliveira Costa, D. do S.; and Rossignolo, J.A. [3], in which a surface probe (25 mm × 25 mm) was used at 10$^{-1}$ °C with a heat transfer analyzer (ISOMET Model 2114, Precision Applied, Bratislava, Slovakia). The chemical composition was characterized using XRF (Shimadzu XRF1800). The analysis and visualization of the microstructure of the pores was performed with an optical microscope (Olympus, 3Z61). The structure of the pores and their intersections between the walls were also visually evaluated [47]. The monitoring of the volumetric expansion of the samples during the burning process was performed by means of camera recordings (SAMSUNG Full HD 27, Samsung, São Paulo, Brazil). The information was stored on a memory card, and was projected in real-time to a television, where images of volumetric expansion were selected during sintering [3,15].

**3. Results and Discussion**

The characterization of glass powder, RHA and CaCO$_3$ were carried out using X-ray fluorescence (Table 1). All materials showed potential to be used in the formulation for the production of vitreous foam, and are suitable for recycling. Glass powder can be added up to 97% by mass of the vitreous foam when using CaCO$_3$ as a foam; this is because CaCO$_3$ is a low-cost binder. The RHA showed a high SiO$_2$ content, favoring its incorporation into

the softened glass mass during low viscosity. The RHA has potential for the production of vitreous foam, because it favored the expansion. The $CaCO_3$ used as a foaming agent was efficient, and had a high concentration of CaO, as it is an industrialized product (Table 1). Loss on fire (LOI) for $CaCO_3$ and RHA are illustrated in Figure 3, where the released $CO_2$ has potential for applications as a porogenic agent.

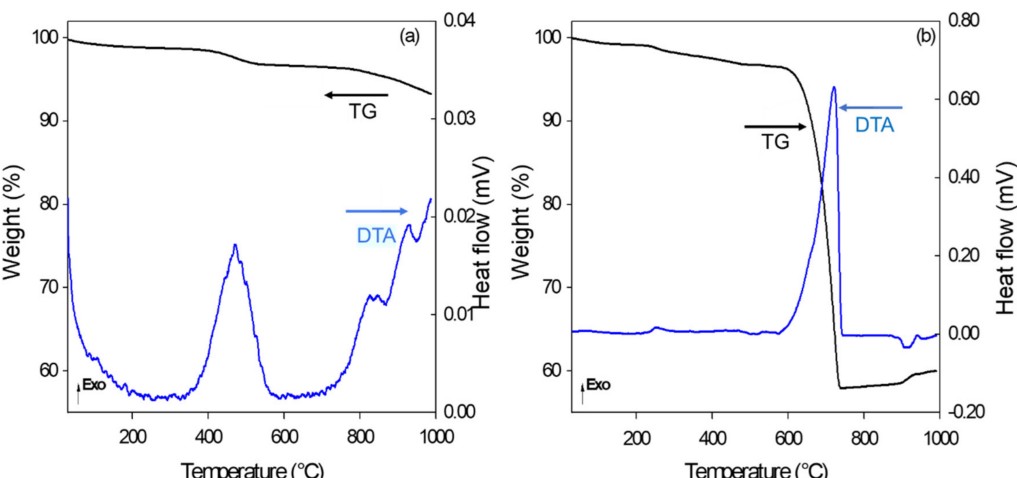

**Figure 3.** Differential thermal (DTA) and thermogravimetric (TGA) analyses of RHA (**a**) and $CaCO_3$ (**b**).

Differential thermal (DTA) and thermogravimetric (TGA) analysis of the porogenic agents are illustrated in Figure 3. The RHA (Figure 3a) presents an endothermic event close to the temperature of 100 °C, which may have been associated with a small mass loss of approximately 1%; this may have been a consequence of residual moisture. Another event was the exothermic peak that occurred at 450 °C, which may have been associated with 2% of the mass loss that was probably related to volatile substances released during burning, such as phosphorus oxide, which is present in RHA, as shown in the chemical analysis [48,49]. Then, two other exothermic events were observed between 800 °C and 850 °C. In this same region, it is possible to observe a mass loss of approximately 4%, which must have been associated with the decomposition of residual carbon into $CO_2$.

The illustrated $CaCO_3$ behavior (Figure 3b) shows an exothermic event close to 700 °C, which occurred due to the loss of mass (±40%) that was very close to the theoretical expected value of 44% by mass for pure $CaCO_3$; this resulted from the release of $CO_2$ [50]. It is important to consider that the determination of the proper burning temperature is essential for the production of glassy foam, as it has a strong relationship with the viscosity of the glass. The volumetric expansion of the vitreous foam is a result of the released gas produced in the decomposition of $CaCO_3$ inside the glass mass; this gas is responsible for the formation of the pore when the viscosity is adequate. The $CaCO_3$ decomposes most efficiently after 660 °C (Figure 3b). The glass depends on higher temperatures to become viscous. This condition guarantees the entrapment of the gas in the softened glass mass, because the viscosity is adequate, contributing to the formation of closed pores, and the production of vitreous foams.

The X-ray diffraction (XRD) results (Figure 4) of the burned samples (750 °C, 800 °C and 850 °C), show an amorphous halo, which resulted from the high concentration of the glass phase. The presence of cristobalite illustrated the existence of the crystalline phase in all analyzed samples ($SiO_2$), and in wollastonite ($CaSiO_3$) [51]. The results show that the RHA reacted in the structure formed by the softened glass powder; the residual carbon burned, which guaranteed the crystallization of the amorphous silica ash in cristobalite. This result was already expected, because the burning of the samples was carried out at temperatures that favor the formation of this silica phase; the peak intensities were compatible with a system with high amorphism [52].

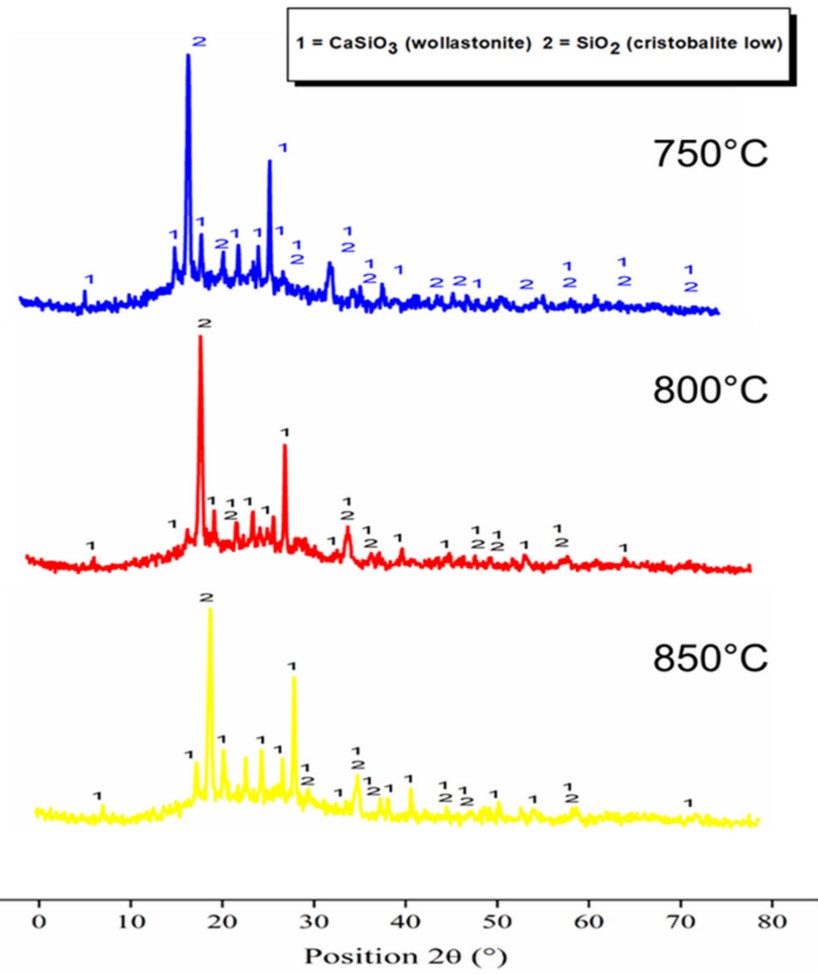

**Figure 4.** XRD patterns of samples sintered at 750 °C, 800 °C and 850 °C.

Figure 5 illustrates the evolution of the volumetric expansion of the foam glass in relation to the gradual increase in temperature (750–850 °C). The observed similarity in geometry of the foam glasses during the sintering process (750 °C and 800 °C) is because for temperature values >850 °C, the glass foam begins to expand, favored by the low viscosity of the sodocalcic glass at these temperatures [53]. The expansion started at the extremities of the matrix, which is a characteristic of the foam glasses (Figure 4c) [54]. The low viscosity presented by the sodocalcic glass at temperatures <850 °C favored the RHA to incorporate into the new cell structure being formed during the vitrification processes [55]. The decomposition of $CaCO_3$ (600 °C to 850 °C) (Figure 5b) that generated $CO_2$ [56] had a strong influence on the foam expansion for temperatures >850 °C, since the volumetric expansion occurred gradually with temperature increase. Figure 6a–cshow the morphology of the foam glasses after sintering (750–800–850 °C).

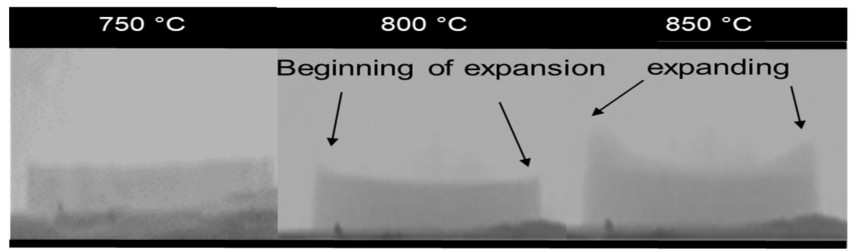

**Figure 5.** Evolution of the volumetric expansion of the foam glass in relation to a gradual increase in temperature at 750 °C, 800 °C and 850 °C.

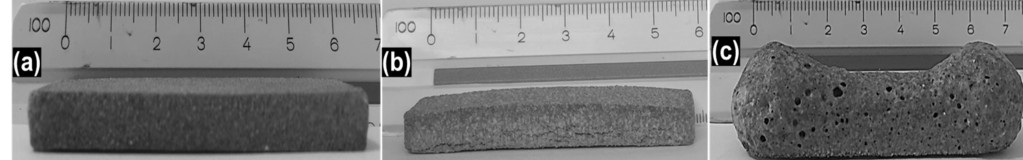

**Figure 6.** Geometric shape of the foam glass sintered at (**a**) 750 °C, (**b**) 800 °C and (**c**) 850 °C.

Figure 6a–c illustrate the morphology and the geometrical shape of the foam glasses after sintering at (a) 750 °C, (b) 800 °C and (c) 850 °C. It was observed that each foam glass presented a distinct geometrical shape. The temperature influenced the sintering of the glass, creating a viscous liquid glass enclosure that contained RHA and $CaCO_3$, forming the new cellular structure illustrated in Figure 6a–c [21,57,58]. The $CaCO_3$ at temperatures <850 °C presented low decomposition, causing a slow rate of gas generation ($CO_2$), so slow that it could not break the walls of the pores created by the housing that formed during sintering; this created a closed system of cavities in the glass (Figure 7) [59]. The formation of pores and cavities on the outer surface of foams resulted from the low pressure that the $CO_2$ exerted at these temperatures on the walls that were forming between the pores and gas during the decomposition of $CaCO_3$. Since the $CO_2$ did not have pressure to exert a tractive force against the pore walls, favoring the escape of the gas inside the cellular structure; only closed pores were formed in all directions of the glass mass composed by glass and RHA [57,60], presenting an average heterogeneous pore size distribution (Figure 7) [61].

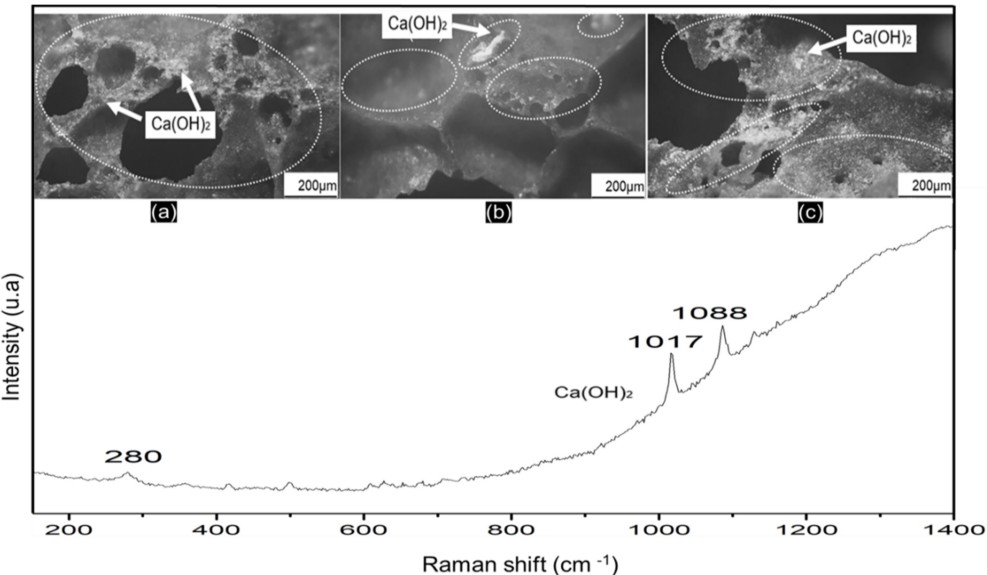

**Figure 7.** Pore morphology with $Ca(OH)_2$ formation in the sintered foams: (**a**) 750 °C, (**b**) 800 °C and (**c**) 850 °C.

The well-defined morphology of the samples and the similarity in addition to the good closed pore concentration (Figure 7a–c) resulted from the glass melting point, which was higher at temperatures 750–850 °C, a condition favored by the addition of RHA [62]. The greater resistance to the melting of the foam glass favored the lower production of open pores [63]. Although measurements of the pore diameter were not performed, the morphology showed (Figure 7a,b) that the smaller pores had a smaller average diameter. One possible explanation is that pores of smaller size create more points of nucleation for the appearance of bubbles, leading to smaller cell diameters [63]. Since the RHA was subjected to a new sintering, it may have presented a typical structure of additivated carbon with excellent properties [64] as a consequence of high amorphous halo (Figure 4a) due to the high content of the glass powder. This condition formed a complex network of fine

pores that contributed to the appearance of a large area of total internal surface, facilitating the incorporation of RHA in the softened glass, and creating the new cellular structure [65].

The $CaCO_3$ behaved as a flux material [19] between the temperatures 750 °C and 850 °C, due to the fact that the heating was slow, causing a high internal shear force, which increased the melting speed of the raw materials, but did not expand [19]. The particle size of the glass ($D_{90}$ = 73 μm) and RHA ($D_{90}$ = 71 μm) contributed to the formation of the new structure, considering that all raw materials had very close particle sizes [62]. This condition favored homogenization and a greater contact surface between glass, RHA and $CaCO_3$, making sintering occur uniformly, increasing the melting speed and the production of closed pores [63]. As these temperatures were low, the viscosity of the molten glass was also low, making it difficult for the gas to expand; thus, there was little change in the volume of the foam glass. This is confirmed because the pores were initially closed pores; after the sample was sintered at high temperature, they tended to become open pores, impacting the apparent density [45]. Figure 7 shows the pore morphology with formation of $Ca(OH)_2$ in the sintered foam glasses (750 °C to 850 °C).

The result of the Raman spectrum (Figure 7) shows that there are peak widths that are due to the overlapping of several Raman peaks. Spectral deconvolution was obtained using Lorentz functions, where three vibrational modes were identified (~280, 1017 and 1088 cm$^{-1}$). These vibrations were in concomitancy with the $CaCO_3$ decomposition [66,67]. For the morphology and formation of $Ca(OH)_2$ in all samples of sintered foam glasses at (a) 750 °C, (b) 800 °C and 850 °C, Raman spectroscopy identified the appearance of calcium hydroxide (Ca $(OH)_2$), with a pH of 12.7 and a density of 2.2 g/cm$^3$. This material was formed by the reaction of the residual water contained and added to the samples with CaO during the exothermic reaction of $CaCO_3$. This statement is verified because industrial $CaCO_3$ presents a high percentage of CaO in its formulation (±97%). The emergence of $Ca(OH)_2$ may have been favored by the $CO_2$ resulting from the oxidation or decomposition reactions of $CaCO_3$ [66], and the carbon remaining in the RHA during the burning process [68]. The formation of pores (Figure 6a–c) showed that there was no similarity in the formation of pores in the foam cell structure depending on the location; pores located inside the foam were spherical in shape, while pores located in the corners were shaped like rounded tetrahedrons, with cavity appearance resulting from the influence of the gradual increase in the sintering temperature (750–850 °C), and the heating rate that influenced the formation of pores [22,69]. Figure 7 shows the bulk density, compressive strength and thermal conductivity of the sintered foam glasses (750–800–850 °C).

Figure 8 shows that the results of sintered foam glasses (750 °C, 800 °C and 850 °C) with heating rate of 30 min indicate apparent densities ranging between 0.2 g/cm$^3$ and 0.29 g/cm$^3$, and thermal conductivities between 0.0026 W/mK and 0.0029 W/mK. The density values are due to low porosity, where density is inversely proportional to porosity [46,51]. Density, mechanical strength and thermal conductivity values are associated with the distribution and morphology (closed pores) produced between 750 °C to 850 °C [42], since these temperatures are close to the sodocalcic glass melting temperature [70], and its viscosity is low, favored by the addition of RHA in the cell structure, as previously mentioned. The apparent densities showed values below 0.30 g/cm$^3$, according to Gibson and Ashby. The foams produced become suitable for use as foam glasses [62,71]. This fact is due to the pores presenting a predominance of closed, non-interconnected pores [58]. Observing Figure 6a–c, it is verified that there were a large number of small pores that influenced the performance and behavior of the foam, in particular the thermal conductivity [62] and resistance to compression [62,72]. The foam glasses produced can be used as thermal insulating material because their thermal conductivity is more than 0.25 W/mK, and they meet the thermal conductivity range of 0.005–0.008 W/mK for newly produced standard commercial foam [33,62,73,74]. The results illustrated in Figure 8 for compressive strength show that with increasing temperature, there is a reduction in mechanical strength. This is a consequence of the action of $CaCO_3$ which decomposed, causing an increase in pores in the foam [57]. The formation of walls between the pores

illustrated in Figure 6a–c shows a well-defined architecture and good thickness, as a result of the incorporation of RHA in the cellular structure of the glass foam, optimized by the low viscosity of soda-calcium glass at these temperatures. This condition also occurred in the foam that received a temperature increase, as observed in Figure 6c, showing the efficiency of the RHA in the formation of the new cellular structure. There may also have existed an influence of the cristobalite present in all foams produced (Figure 4), since it appears that cristobalite influences porosity and the mechanical resistance of foam glass [58]. The values presented for compressive strength in all samples can also be related to the effect of increasing the degree of adhesion between the sodocalcic glass, RHA and $CaCO_3$ interfaces [15]. The vitreous foam sintered at 850 °C showed a loss of compressive strength (17%); this loss of strength was expected as a result of the decrease in apparent density at this stage of 0.24 g/$cm^3$. Moreover, at this temperature, the compressive strength reduced by approximately 35%. The loss of compressive strength may have occurred due to the rupture of the pore walls; therefore, the leakage of $CO_2$, which is a consequence of the high internal pressure that expands the glass foam due to the low viscosity that existed, caused the foam to exhibit an inverse mechanical strength behavior [38]. The compressive strength values presented by the foams investigated in this study met the standard value in the literature for compressive strength of commercial glass foam, ranging from 0.7 to 2.5 MPa [42,53]. All manufactured foams show a good positive correlation between apparent density, thermal conductivity and compressive strength, showing potential for use as thermal insulation panels due to their thermal conductivities below 0.08 W/mK [58].

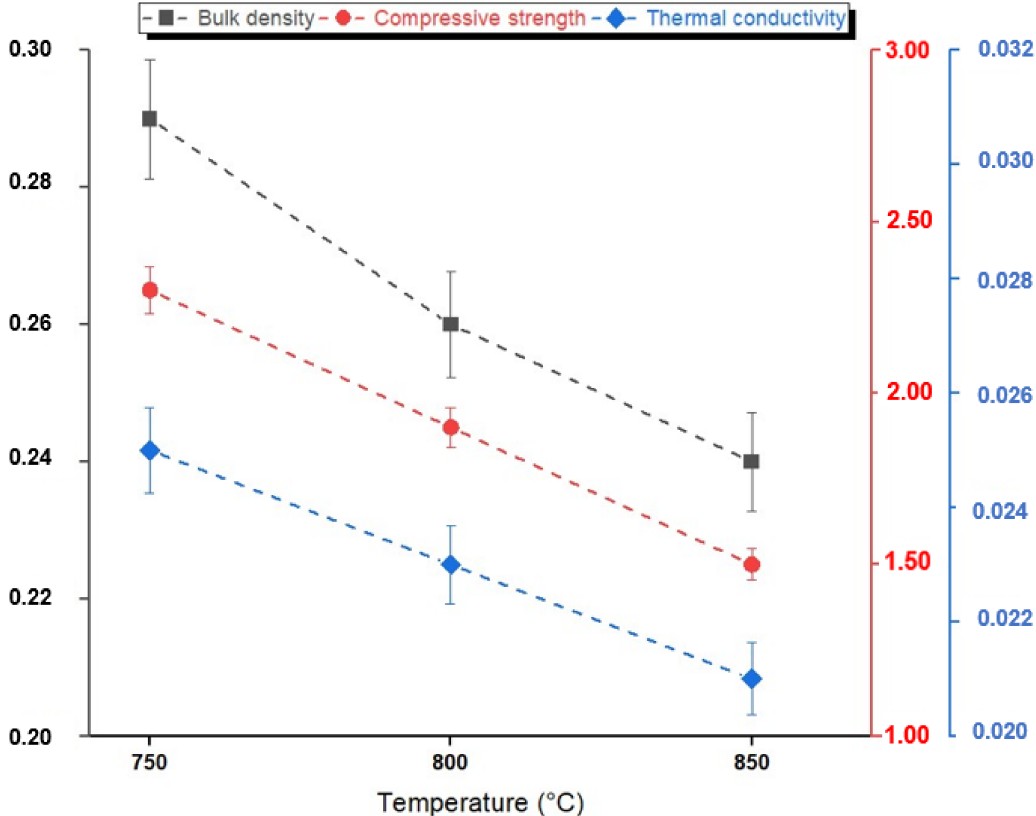

**Figure 8.** Bulk density, compressive strength and thermal conductivity of the sintered foam glasses (750–800–850 °C).

## 4. Conclusions

This research successfully prepared and characterized foam glasses with technological properties that meet the needs of the construction industry in relation to materials with thermal and mechanical insulating properties. The use of residues such as glass powder from beverage bottles and rice husk ash, using calcium carbonate as a foaming agent, makes its production feasible. Burning at lower temperatures (750–800–850 °C) makes the production of this material economically viable. The following conclusions were drawn within the limits of this study:

- Glasses of discarded beverage bottles can be fully recycled as raw material for the production of foam glasses with excellent mechanical properties (0.7 to 2.5 MPa) and thermal conductivities (0.0026 W/mK to 0.0029 W/mK).
- The sintering temperature has a strong influence on the properties of the foam glass. The temperatures 750–800–850 °C are ideal for producing closed-pore vitreous foams with technological properties that serve the civil construction industry.
- The resulting microstructure is strongly influenced by rice house ash and glass, where $CaCO_3$ has excellent performance as a foaming agent during firing (750–800–850 °C).
- The particle size of the glass powder and rice husk ash contribute to the formation of the new cell structure and ensures greater homogenization of the raw materials, causing the glass mass to incorporate the rice husk ash mass during the period in which the glass is softened.
- In terms of physical properties, the density, thermal conductivity and resistance to compression decrease with increasing temperature, meeting the requirements for commercial glass foams. Porosity has the opposite behavior.
- The morphology of the foam glass manufactured with the addition of sodocalcic glass (78%wt), rice husk ash (16%wt) and calcium carbonate (6%wt) as foaming agent sintered at 750–800–850 °C had most pores closed, with well-defined architecture and walls with good thickness, ensuring the enclosure of $CO_2$ within the cell structure of the foam glass.
- The sodocalcic glass favored the viscosity, presenting a higher value (0.29 g/cm$^3$) when sintered at 750 °C, and 0.26 g/cm$^3$ at 850 °C. These results were made possible by the influence of the addition of rice husk ash in the foam glass matrix.
- Low density is one of the most important properties for the production of foam glasses, because it influences the thermal conductivity and resistance to compression.
- All ideal processing parameters for the production of foam glasses for thermal insulation were obtained in all foams produced in this study. The sintered glassy foam (750 °C) showed porosity (83%) below the standard for commercial foam glass (>85%), but presented the best result for compressive strength (2.72 MPa) and thermal conductivity within the limits (0.0029 W/mK) for the same temperature.
- Burning at lower temperatures could be investigated to identify the behavior of the properties investigated in this study.
- An investigation into the type of gas produced during the new burning at temperatures 750–800–850 °C could bring very important information about the type of gas produced in the burning.
- New residues could be added together with rice husk ash and glass powder to study the behavior of the material.

**Author Contributions:** F.A.d.S.F., Martin, C.A.G.M. and D.d.S.d.O.C.; formal analysis, F.A.d.S.F., Martin, C.A.G.M. and J.A.R.; investigation, F.A.d.S.F., D.d.S.d.O.C. and J.A.R.; methodology, F.A.d.S.F., D.d.S.d.O.C. and J.A.R.; resources, F.A.d.S.F., D.d.S.d.O.C. and J.A.R.; writing—original draft preparation, F.A.d.S.F., D.d.S.d.O.C. and J.A.R.; writing—review and editing, F.A.d.S.F., D.d.S.d.O.C. and J.A.R.; visualization, F.A.d.S.F., D.d.S.d.O.C. and J.A.R. All authors have read and agreed to the published version of the manuscript.

**Funding:** This research received no external funding.

**Institutional Review Board Statement:** Not applicable.

**Informed Consent Statement:** Not applicable.

**Data Availability Statement:** Data available on request.

**Acknowledgments:** This research was supported for publication by PROPESP/UFPA (PAPQ).

**Conflicts of Interest:** The authors declare no conflict of interest.

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
