# Peer review of "Vitreous Foam with Thermal Insulating Property Produced with the Addition of Waste Glass Powder and Rice Husk Ash"

_sustainability, doi:10.3390/su15010796_

Round 1

Reviewer 1 Report (Previous Reviewer 1)

The paper generally is good but it must be improved before acceptance. The following should be completed.

Important results should be added into abstract in terms of %.

The authors should mention the use of waste organic materials in the introduction. Following studies should be added for this purpose:  composition component influence on concrete properties with the additive of rubber tree seed shells; normal-weight concrete with improved stress–strain characteristics reinforced with dispersed coconut fibers

Almost no information was inclded for use of waste glass powder in the introduction. The following should be added to introduce waste glass powder: influence of replacing cement with waste glass on mechanical properties of concrete; concrete containing waste glass as an environmentally friendly aggregate: a review on fresh and mechanical characteristics; mechanical behavior of crushed waste glass as replacement of aggregates; flexural behavior of reinforced concrete beams using waste marble powder towards application of sustainable concrete

Introduction section is not enough. There are hundreds of studies for these subjects. Please expand the introduction as much as possible. 

The reason for chosing this materials to combine should be explained.

Add photos of used materials and samples in material method section.

Add photos of test setup

Add photos of samples before and after tests

Future needs and recomendations should be added to conclusion.

Author Response

Dear reviewer
Attached is the letter dealing with the guidelines for improving the work.

Reviewer 2 Report (Previous Reviewer 2)

Please format correctly the citations marked in red on pages 2 and 5 and insert the same in the Reference section.

Author Response

Dear reviewer
Attached is the letter dealing with the guidelines for improving the work.

Round 2

Reviewer 1 Report (Previous Reviewer 1)

The paper can be accepted.

This manuscript is a resubmission of an earlier submission. The following is a list of the peer review reports and author responses from that submission.

Round 1

Reviewer 1 Report

In the abstract important outcomes of the study should be given.

Starting 1st sentence of the introduction with "on the other hand" is not proper. Please combine first two pragraph and extend it with importance of use of waste materials. The following articles must be used for this purpose: use of recycled coal bottom ash in reinforced concrete beams as replacement for aggregate; flexural behavior of reinforced concrete beams using waste marble powder towards application of sustainable concrete; improvement in bending performance of reinforced concrete beams produced with waste lathe scraps; performance assessment of fiber-reinforced concrete produced with waste lathe fibers; performance evaluation of fiber-reinforced concretes produced with steel fibers extracted from waste tire; performance evaluation of fiber-reinforced concretes produced with steel fibers extracted from waste tire

Since introduction is limited, novelty of this study is not clear. Please add novelty of the study in a paragraph

Introduction section is not enough. There are hundreds of studies for these subjects. Please expand the introduction as much as possible. The following must be add for use of glass recycling: influence of replacing cement with waste glass on mechanical properties of concrete; concrete containing waste glass as an environmentally friendly aggregate: a review on fresh and mechanical characteristics; mechanical behavior of crushed waste glass as replacement of aggregates

The reason for chosing this materials to combine should be explained.

Add photos of used materials and samples in material method section.

Add photos of test setup

Add photos of samples before and after tests

Future needs and recomendations should be added to conclusion.

Author Response

Mr. Reviewer, please find attached answer

Reviewer 2 Report

Vitreous foam with thermal insulating property produced with the addition of waste glass powder and rice husk ash 

Some points must be better improved in a manuscript for possible publication.

Introduction

It’s seems that missing an initial phrase in first paragraph on introduction.

p. 2, ln 47-83. This paragraph is too long, with many subjects that are presented. The same can be presented in different paragraphs, showing the results of previous investigations separately in all manuscript. I suggest that author includes these references about use of different materials considering circular economy and sustainability topics:

https://doi.org/10.3390/su14116740

https://doi.org/10.1016/j.jclepro.2014.02.040

https://doi.org/10.1016/j.cemconres.2014.01.015

https://doi.org/10.1007/s10163-022-01493-8

The manuscript's originality must be presented at the end of the introduction.

This section must be significantly improved.

Materials and methods

This section must be divided accordingly. The first item is “Material preparation”, without present what materials will be processed. References like “(TS RUBENS, Brazil)” and “(XRF, 1800, Shimadzu, São Paulo, Brazil)” must be rewritten. The CaCO3 properties and classification according Brazilian standards must be presented in a detailed form.

Experimental program must be presented in a direct way, using tables to sumarize the investigations performed, number and samples dimensions. 

Figure 6 replace the comma by dots.

An extensive English edition is necessary

Author Response

(The authors gave the same response as above.)

Reviewer 3 Report

After reviewing the paper entitled “Vitreous foam with thermal insulating property produced with the addition of waste glass powder and rice husk ash” I decided to ask for mayor revisión, despite the fact that authors have not taken care both in writing and editing the paper, but it has some relevance for scientific knolowledge.

Introduction begins with: “On the other hand”. Where is the “one hand”?.

Self-reference authors in introduction is irrelevant. Moreover there are some interesting researchs that are not included in the state of the art. Authors should improve the state of the art, including some aspects such as the influence of the temperatura in the glass foam forming (but not including only one paragraph).

Authors should improve materials description, also including PVA, and methods, including Raman test.

The references of the standards used should be cited in the text.

Results and discussion should be strongly improved. Authors repeat some paragraphs and discusión (p.e lines 150 to 160, 212 to 222). Moreover, they mistakenly refer to figure 6 in line 233. Lines 297 to 302 are in portuguese…

Also, authors should take care with the subscripts in chemical compounds.

Finally, conclusions should be rewriten for its strong improvement according to the objetives and results obtained.

Author Response

(The authors gave the same response as above.)

Round 2

Reviewer 1 Report

The authors did not complete any of the request.

Please complete each question in the first revision.

Q8 is not answered. I could not see any modifications for future needs and recommendations in conclusion. These are very important aspects for both readers (as engineers or manufacturers) and also researchers.   Q5, Q6 and Q7: "Dear reviewer, we ask you to reconsider your request because the inclusion of photos will not contribute to the results of this study." This is not acceptable answer to reviwer. The photos are cruciual for the readers and also for the researches who wants to conduct smiliar studies. Include them all.   Q2, Q3 and Q4 are answered based on the request of only Review 2. Check the request of Review 1. The added ones are also not relavent with thermal properties. Include Reviewer 1 requests too.   The results of the reviewer are not adequate and acceptable for this prestigous journal. The authors should be answered all questions with care and detail.

Author Response

Dear reviewer, thank you for your suggestions to enhance the work "Vitreous foam with thermal insulating property produced with the addition of waste glass powder and rice husk ash", in order to make it more objective for readers of this prestigious magazine. We apologize for not responding to all items in the previous review. Thank you very much !

Reviewer 2 Report

Some suggestions made at the first round don't incorporate in this second version. According to the authors, the problem is related to the worldwide pandemic. However, in my opinion, this point is insufficient to justify the lack of information in some points of the manuscript. In this way, I strongly suggest that the authors reply and complete the appointments made previously for a possible publication.  

Author Response

(The authors gave the same response as above.)

Reviewer 3 Report

After the second review of the paper entitled “Vitreous foam with thermal insulating property produced with the addition of waste glass powder and rice husk ash” I decided to REJECT it because authors have practically disregarded the recommendatios I made before.

Author Response

(The authors gave the same response as above.)
